# Skills Regularized Task Decomposition for Multi-task Offline Reinforcement Learning

**Minjong Yoo, Sangwoo Cho, Honguk Woo**[*]
Department of Computer Science and Engineering
Sungkyunkwan University
{mjyoo2, jsw7460, hwoo}@skku.edu

## Abstract

Reinforcement learning (RL) with diverse offline datasets can have the advantage of leveraging the relation of multiple tasks and the common skills learned across those tasks, hence allowing us to deal with real-world complex problems efficiently in a data-driven way. In offline RL where only offline data is used and online interaction with the environment is restricted, it is yet difficult to achieve the optimal policy for multiple tasks, especially when the data quality varies for the tasks. In this paper, we present a skill-based multi-task RL technique on heterogeneous datasets that are generated by behavior policies of different quality. To learn the shareable knowledge across those datasets effectively, we employ a task decomposition method for which common skills are jointly learned and used as guidance to reformulate a task in shared and achievable subtasks. In this joint learning, we use Wasserstein auto-encoder (WAE) to represent both skills and tasks on the same latent space and use the quality-weighted loss as a regularization term to induce tasks to be decomposed into subtasks that are more consistent with high-quality skills than others. To improve the performance of offline RL agents learned on the latent space, we also augment datasets with imaginary trajectories relevant to high-quality skills for each task. Through experiments, we show that our multi-task offline RL approach is robust to the mixed configurations of different-quality datasets and it outperforms other state-of-the-art algorithms for several robotic manipulation tasks and drone navigation tasks.

## 1 Introduction

In the reinforcement learning (RL) field, the offline RL research has recently gained much attention, as numerous works showed that it is effective for various problems of sequential decision making to adopt a data-driven learning mechanism using previously collected data of experiences and trajectories [1, 2, 3]. In the meanwhile, multi-task RL is considered promising to enhance the generality of RL policies and improve the learning efficiency [4, 5, 6, 7, 8, 9, 10]. Recently, a data sharing method for multi-task learning was introduced to address the issue of limited data for real-world control applications [11]. Yet, multi-task RL has not been fully investigated in offline settings.

In the offline RL context, we present a novel multi-task model by which a single policy for multiple tasks can be data-efficiently achieved and its learning procedure is robust to heterogeneous datasets of different quality. In offline RL where interaction with the environment is not allowed and arbitrary or low-performance behavior policies might be involved in data collection, it is important to maintain the robustness in learning on different-quality data. To this end, we devise a joint learning mechanism of skill (short-term action sequences from the datasets) and task representation, which enables the task decomposition into achievable subtasks via quality-aware skill regularization. We also employ

---

[*]Honguk Woo is the corresponding author.

36th Conference on Neural Information Processing Systems (NeurIPS 2022).

data augmentation based on high-quality skills, thus creating plausible trajectories and alleviating the limited quality and scale issues of offline datasets. Through experiments, we demonstrate that our model achieves robust performance for multi-task robotic manipulation and drone navigation, without requiring additional interaction with the environment.

The main contributions of this paper are summarized as follows.

- We present a novel multi-task offline RL model that enables the task decomposition into achievable subtasks through the quality-aware joint learning on skills and tasks. The model ensures the robustness of learned policies upon the mixed configurations of different-quality datasets.
- We devise the data augmentation scheme specific to multi-task RL on limited offline datasets, aiming at creating imaginary trajectories that are likely to be generated by expert policies.
- We evaluate our model under multi-task robot and drone scenarios and demonstrate its benefit particularly upon heterogeneous data conditions.

## 2 Overall Approach

In this section, we describe the problem of making optimal decisions upon a multi-task Markov decision process (MDP) and briefly present our data-driven approach to the problem.

### 2.1 Preliminary

Reinforcement learning (RL) offers an active learning framework based on MDPs for tackling sequential decision problems. In conventional RL formulation, a learning environment is represented as an MDP $\mathcal{M}$ with $(\mathcal{S}, \mathcal{A}, \mathcal{P}, R, \gamma)$ where $\mathcal{S}$ is a state space, $\mathcal{A}$ is an action space, $\mathcal{P}(s_{t+1}|s_t, a_t)$ is a transition probability for states $s_t, s_{t+1} \in \mathcal{S}$ and an action $a_t \in \mathcal{A}$, $R(s_t, a_t)$ is a reward function, and $\gamma \in [0, 1]$ is a discount factor. The objective of an RL agent is to find an optimal policy $\pi^*(a|s)$ by which the cumulative discounted reward $J(\pi) = \mathbb{E}_\pi[\sum_{t=0}^\infty \gamma^t r_t]$ [12] is maximized. In general, the policy $\pi$ is learned via continual interaction with a learning environment defined in an MDP.

**Offline RL** Offline RL aims at maximizing the cumulative discounted reward $J(\pi)$ as explained above within the same MDP formulation; however, unlike conventional RL, it is assumed to use only static datasets of previously collected trajectories $\mathcal{D} = \{(s_t, a_t, r_t, s_{t+1})\}_t$ for training. Likewise, it rarely considers direct interaction with the environment. Offline RL algorithms can increase the usability of previously collected data in the domain of making sequential decisions where temporal credit assignment with long time horizons is important.

**Multi-task RL** Multi-task RL considers more than a single task when achieving the optimal policy $\pi^*$. It is normally formulated as a family of MDPs $\{\mathcal{T}_i = (\mathcal{S}, \mathcal{A}, \mathcal{P}_i, R_i, \gamma)\}_i$ where each individual task $\mathcal{T}_i$ is associated with its respective MDP and it is sampled according to a task distribution $p(\mathcal{T})$.

**Hidden Parameter MDP** To represent the implicit temporal dynamics properties in a multi-task environment, which are relevant to the Markovian properties of each task, we introduce a hidden latent variable $v_t$ [13]. That is, we have $R(s_t, v_t, a_t) := R_{v_t}(s_t, a_t)$ and $\mathcal{P}(s_{t+1}, v_{t+1}|a_t, s_t, v_k) := \mathcal{P}_{v_t}(s_{t+1}|a_t, s_t)$ for $s_t, \ s_{t+1} \in \mathcal{S}$ and $a_t \in \mathcal{A}$, where the actual state space is extended to $\mathcal{S} \times \mathcal{V}$ and $\mathcal{V}$ is the collection of latent variables $v_t$. Then, we obtain the respective partially observable MDP (POMDP) that is formulated as a tuple $(\mathcal{S} \times \mathcal{V}, \ \mathcal{A}, \ \Omega, \ \mathcal{P}_\mathcal{V}, \ R_\mathcal{V}, \ \mathcal{O}, \ \gamma)$ where $\Omega = \mathcal{S}$ and $\mathcal{O}(s_t, v_t) \mapsto s_t$ denote the observation space and the observation function, respectively.

### 2.2 Overall Approach For Multi-task Offline RL

In principle, offline RL algorithms train a policy $\pi$ by optimizing the objective $\text{argmax}_\pi J_D(\pi) - \alpha \cdot c(\pi, \pi_\mathcal{D})$ which tends to mitigate the extrapolation problem [1]. $J_\mathcal{D}$ is the average return of the learned policy $\pi$ in the empirical MDP $\tilde{\mathcal{M}}$ derived from static datasets $\mathcal{D}$, which are generated by a behavior policy $\pi_\mathcal{D}$. Some cost function $c(\cdot)$ is used to regulate the distance of the policies $\pi$ and $\pi_\mathcal{D}$ and $\alpha$ is a hyperparameter. With this regularization by the behavior policy, offline RL algorithms are often vulnerable to low-quality datasets. Overfitting problems can occur such that the maximum average return $\max_\pi J_\mathcal{D}(\pi)$ of $\tilde{\mathcal{M}}$ is much lower than that of its respective true MDP $\mathcal{M}$, when a low-performance or arbitrary policy is used for data generation.

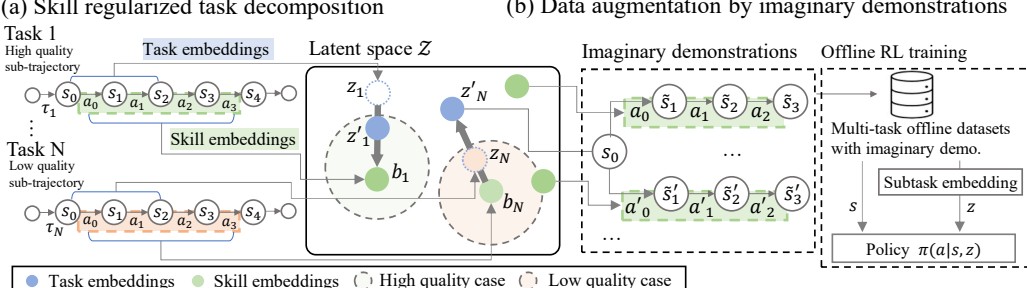

(a) Skill regularized task decomposition

(b) Data augmentation by imaginary demonstrations

Figure 1: Our proposed multi-task offline RL model consisting of (a) task decomposition and (b) data augmentation. In (a), sub-trajectories from static datasets are converted into skill embeddings and task embeddings on the same latent space, which together enable the decomposition of tasks into achievable subtasks. The blue-colored dots denote task embeddings that model the environment, and the green-colored dots denote skill embeddings. In the green-colored dotted circle, a sub-trajectory $\tau_1$ of task 1 is embedded as $z_1$ and then located as $z_1'$ closer to its corresponding high-quality skill $b_1$ (the action sequence of the sub-trajectory $\tau_1$ with large returns), while in the red-colored dotted circle, another sub-trajectory $\tau_N$ of task $N$ is embedded as $z_N$ and located as $z_N'$ further from its corresponding low-quality skill $b_N$ (the action sequence of the sub-trajectory $\tau_N$ with small returns). In (b), for training offline RL agents, imaginary trajectories similar to expert demonstrations are sampled from the latent space and added to the datasets.

In multi-task offline RL, we reformulate a family of MDPs $\{\mathcal{T}_i\}_i$ as a hidden parameter MDP in that multiple MDPs are combined into a single POMDP based on hidden parameters that specify temporal Markovian properties of the environment. While the overfitting issue of offline RL can be alleviated by exploring the relation of multiple tasks and inducing the shareable knowledge from their datasets in a multi-task setting, it is not guaranteed that inferring the hidden parameters fully enables the well-structured representation of related tasks. It is because the behavior policy heterogeneity and state-action pair disparity of tasks can prevent the sub-trajectories of common-knowledge tasks from being closely mapped on the latent space [14, 15].

To address the issue of different-quality and heterogeneous datasets in the context of multi-task offline RL, we take a novel task embedding approach that combines (a) skill-regularized task decomposition and (b) data augmentation by imaginary demonstrations, as illustrated in Figure 1. (a) The former enables the decomposition and reformulation of an individual task in the latent space of achievable subtasks by jointly learning the common skills and adapting the subtasks in more achievable representation via quality-aware skill regularization (in Section 3). This task decomposition establishes subtask embeddings on the same latent space in which agents can be efficiently trained. (b) The latter improves the performance of offline RL agents trained on the latent space by augmenting datasets with imaginary demonstrations (in Section 4). For each task, its well-matched skills are consequently used to generate additional trajectories similar to expert data without online interaction. This can improve the adaptability of offline RL algorithms for such tasks that have only low-quality data or do not exist in the static datasets. Then, using the subtask decomposition and augmented datasets with imaginary demonstrations, a multi-task policy is learned via offline RL algorithms.

## 3 Task Decomposition with Quality-aware Skill Regularization

In this section, we describe the task decomposition model with quality-aware skill regularization on multi-task offline datasets. The model includes skill embedding and task embedding networks that are jointly learned with a regularization term based on the behavior quality, as shown in Figure 2. Specifically, we define short-term action sequences as skills and embed them in a latent space. By jointly learning skill embeddings and task embeddings, we induce tasks to be transformed into subtasks that are achievable by and aligned with skills. In doing so, we incorporate the behavior quality of each sub-trajectory into the joint learning procedure as part of skill regularization, thus mitigating the adverse effect by low-quality data.

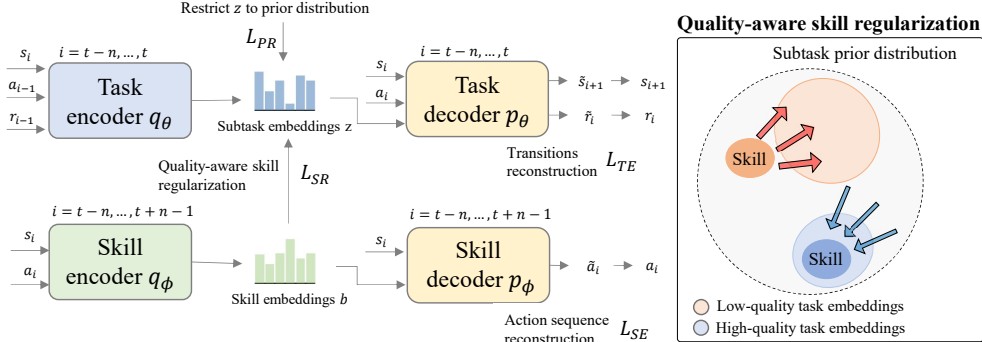

Figure 2: Task decomposition procedure with quality-aware skill regularization. In the right side of the figure, the red arrow denotes $L_{PR}$ in (2) that makes low-quality sub-trajectories stretch within the prior distribution of tasks (in gray), and the blue arrow denotes $L_{SR}$ in (4) that makes high-quality sub-trajectories shrink around the distribution of skills (in blue).

## 3.1 Learning Skill Embeddings

To represent the agent's behavior to a latent space $\mathcal{Z}$, we use an auto-encoder mechanism. An encoder $q_\phi$ takes as input a sequence of state-action pairs $d_t = (s, a)_{t-n:t+n-1}$ for time interval $[t-n, t+n-1]$, mapping it to a latent vector $b_t \in \mathcal{Z}$, while a decoder $p_\phi$ reconstructs the input action sequence $a_{t-n:t+n-1}$ from the pair of $b_t$ and $s_{t-n:t+n-1}$. We term the latent vector $b_t$ *skill embeddings*, considering that action sequences on short-term horizons capture agent's behaviors for a specific task. For maintaining the learning stability on skill embeddings $b_t \in \mathcal{Z}$, we use Wasserstein auto-encoder (WAE) [16] with the maximum mean discrepancy (MMD)-based penalty and a prior distribution on $b_t$. Then, the loss function is defined as

$$L_{SE}(\phi) = \frac{1}{m} \sum_{i=1}^{m} \sum_{j=-n}^{n-1} \|a_{t_i+j} - p_\phi(s_{t_i+j}, q_\phi(d_{t_i}))\|_2 + \lambda \cdot L_{PR}(\{\tilde{b}_i\}_{i=1}^{m}, q_\phi(\{d_{t_i}\}_{i=1}^{m})) \quad (1)$$

where $\{\tilde{b}_i\}_{i=1}^{m} \sim P_B$ is sampled by a prior of skill embeddings, $\lambda > 0$ is a prior distribution-based regulation hyperparameter, and $\{s_{t_i}, a_{t_i}, d_{t_i}\}_{i=1}^{m} \sim \mathcal{D}$. Here, $L_{PR}$ is used to restrict skill embeddings,

$$L_{PR}(\{b\}, \{\tilde{b}\}) = \frac{1}{m(m-1)} \sum_{i \neq j} k(b_i, b_j) + \frac{1}{m(m-1)} \sum_{i \neq j} k(\tilde{b}_i, \tilde{b}_j) - \frac{1}{m^2} \sum_{i,j} k(b_i, \tilde{b}_j) \quad (2)$$

where $m$ is the size of $\{b\}, \{\tilde{b}\}$ and $k : \mathcal{Z} \times \mathcal{Z} \to \mathbb{R}$ is the positive-definite reproducing kernel.

## 3.2 Skill-regularized Task Decomposition

To decompose an individual task into a set of shared and achievable subtasks, we use skill embeddings as guidance for the analysis of relation between tasks. We first view each task as a composition of subtasks which can be modeled as a hidden parameter MDP. For task embeddings, we then use the WAE-based model architecture similar to skill embeddings previously described. For sub-trajectories $\tau_t = (s_{t-n:t}, a_{t-n-1:t-1}, r_{t-n-1:t-1})$ of $n$-length transitions each, we have an encoder $q_\theta : \tau_t \longmapsto z_t \in \mathcal{Z}$ to yield task embeddings and a decoder $p_\theta : (s_t, a_t, z_t) \longmapsto (s_{t+1}, r_t)$ to express the transition probability $\mathcal{P}$ and reward function $R$. Then, for $\{s_{t_i}, a_{t_i}, \tau_{t_i}\}_{i=1}^{m} \sim D$, the loss function for task embeddings is defined as

$$L_{TE}(\theta) = \frac{1}{m} \sum_{i=1}^{m} \sum_{j=-n}^{0} \|(s_{t_i+j+1}, r_{t_i+j}) - p_\theta(s_{t_i+j}, a_{t_i+j}, q_\theta(\tau_{t_i}))\|_2. \quad (3)$$

We also add a skill regularization term. Since in offline RL datasets, trajectories are not necessarily generated by experts or optimal policies for all tasks, we incorporate the quality of trajectories in

---

**Algorithm 1** Skill regularized task decomposition

---

Offline dataset $\mathcal{D}$, subtask embedding parameter $\theta$, skill embedding parameter $\phi$

Regulation hyperparameter $\lambda$, batch size $m$, learning rate $\eta$

**loop**

    Sample $\{d_{t_i}, \tau_{t_i}, s_{t_i-n:t_i+n}, a_{t_i-n:t_i+n}, r_{t_i-n:t_i}\}_{i=1}^m \sim \mathcal{D}$

    $\{b_{t_1}, b_{t_2}, ..., b_{t_m}\} = q_\phi(\{d_{t_1}, d_{t_2}, ..., d_{t_m}\}), \{z_{t_1}, z_{t_2}, ..., z_{t_m}\} = q_\theta(\{\tau_{t_1}, \tau_{t_2}, ..., \tau_{t_m}\})$

    $\{\tilde{b}_0, \tilde{b}_1, ..., \tilde{b}_n\} \sim P_B = \mathcal{N}(0, 1), \ \ \{\tilde{z}_0, \tilde{z}_1, ..., \tilde{z}_n\} \sim P_Z = \mathcal{N}(0, 1)$

    $L_{SE}(\phi) = \frac{1}{m}\sum_{i=1}^m \sum_{j=-n}^{n-1} \|a_{t_i+j} - p_\phi(s_{t_i+j}, b_{t_i})\|_2 + L_{PR}(\{b_{t_i}\}_{i=1}^m, \{\tilde{b}_i\}_{i=1}^m)$ using (1)

    $L_{TE}(\theta) = \frac{1}{m}\sum_{i=1}^m \sum_{j=-n}^{0} \|(s_{t_i+j+1}, r_{t_i+j}) - p_\theta(s_{t_i+j}, a_{t_i+j}, z_{t_i})\|_2$ using (3)

    $L_{SR}(\theta) = \frac{1}{m}\sum_{i=1}^m \tilde{R}(s_{t_i}, a_{t_i}) \cdot \|q_\theta(\tau_{t_i}) - q_\phi(d_{t_i})\|_2$ using (4)

    $L_{SRTD}(\theta) = L_{TE}(\theta) + L_{PR}(\{z_{t_i}\}_{i=1}^m, \{\tilde{z}_i\}_{i=1}^m)) + L_{SR}(\theta)$ using (5)

    $\phi \leftarrow \phi + \eta \cdot \nabla L_{SE}, \ \theta \leftarrow \theta + \eta \cdot \nabla L_{SRTD}$

**end loop**

**return** $\theta, \phi$

---

the regularization term. Once sub-trajectories are transformed in skill embeddings, their quality is estimated by the episodic returns $\tilde{R}(s, a)$. The quality-aware skill regularization loss is defined as

$$L_{SR}(\theta) = \frac{1}{m}\sum_{i=1}^m \tilde{R}(s_{t_i}, a_{t_i}) \cdot \|q_\theta(\tau_{t_i}) - q_\phi(d_{t_i})\|_2. \tag{4}$$

The overall loss is defined as

$$L_{SRTD}(\theta) = L_{TE}(\theta) + L_{SR}(\theta) + \lambda L_{PR}(\{\tilde{z}_i\}_{i=1}^m \sim P_Z, q_\theta(\{\tau_{t_i}\}_{i=1}^m)) \tag{5}$$

where $P_Z$ is a prior of task embeddings. This allows the encoder $q_\theta$ to generate the embeddings at subtask-level (or subtask embeddings) upon a sequence of sub-trajectories which are agnostic to tasks in multi-task settings. In particular, each task is represented within close proximity to some high-quality skills learned from the trajectories with large episodic returns. By increasing the use of high-quality skills task-agnostically, this task decomposition reduces the adverse effect of low-quality data and induces the task decomposition into more achievable subtasks. When training offline RL agents, we use the output of $q_\theta$ as part of the state input to RL algorithms. Algorithm 1 implements the learning procedure explained in (1)-(5).

Here we provide the analysis of skill regularization effects in (4). Let $q$ and $p$ be a skill encoder and decoder obtained by minimizing the loss in (1), and consider $p$ as part of the environment similarly in [17, 18]. Then, the decoder $p$ follows the MDP $\mathcal{M}^p = (\mathcal{S}, \mathcal{A} = \mathcal{Z}, \mathcal{P}^p, \mathcal{R}^p, \gamma)$ in which a high-level (skill) action $z_t \in \mathcal{Z}$ is converted to such a low-level (primitive) action $a_t \sim p(\cdot|s_t, z_t)$ that directly interacts with the environment. Furthermore, assume that sub-trajectories $\tau$ in (3) and a sequence of state-action pairs $d$ in (1) are restricted to the current state. Then, we obtain such high-level policies $q_\theta$ and $q$ which are trained for the MDP $\mathcal{M}^p$. As the output of $q_\theta$ is contained in the input states of $\mathcal{M}^p$, our objective is to maximize the performance gap between $q_\theta$ and $q$; i.e., maximize $\eta(\theta) := J_p(q_\theta) - J_p(q)$ where $J_p$ is the average return in $\mathcal{M}^p$. Following [19], we obtain that $\eta(\theta) = \mathbb{E}_{s \sim d_{q_\theta}, z \sim q_\theta}[\mathcal{R}_{s,z}^q - V^q(s)]$ where $d_{q_\theta}$ is a state visit distribution induced by $q_\theta$, $\mathcal{R}_{s,z}^q$ is an episodic return induced by $q$, and $V^q$ is a value function of $q$. However, in offline RL, it is difficult to approximate $q_\theta$ precisely, so we rather want to use the distribution of $q$ for the state visitation distribution of $q_\theta$ without much propagation error. To do this, we optimize $\tilde{\eta}(\theta) = \mathbb{E}_{s \sim q, z \sim q_\theta}[\mathcal{R}_{s,z}^q - V^q(s)]$ under the restriction that $q$ and $q_\theta$ remain in close proximity [20],

$$\text{maximize}_\theta \ \tilde{\eta}(\theta) := \mathbb{E}_{s \sim d_q, z \sim q_\theta}[\mathcal{R}_{s,z}^q - V^q(s)] \text{ subject to } KL(q_\theta(\cdot|s)||q(\cdot|s)) \leq \epsilon. \tag{6}$$

Under the KL-divergence constraint in (6), we obtain that the state distribution of $d_q$ corresponds to that of the dataset $\mathcal{D}$, implying $s \sim d_q \cong \mathcal{D}$.

Using the classical Lagrangian argument, we can simplify the optimization problem in (6) into a nonconstraint optimization problem,

$$L_{\text{SR}}(\theta) = \mathbb{E}_{s \sim q, z \sim q_\theta}[\mathcal{R}_{s,z}^q - V^q(s)] + \beta(\epsilon - KL(q_\theta(\cdot|s)||q(\cdot|s)) \tag{7}$$

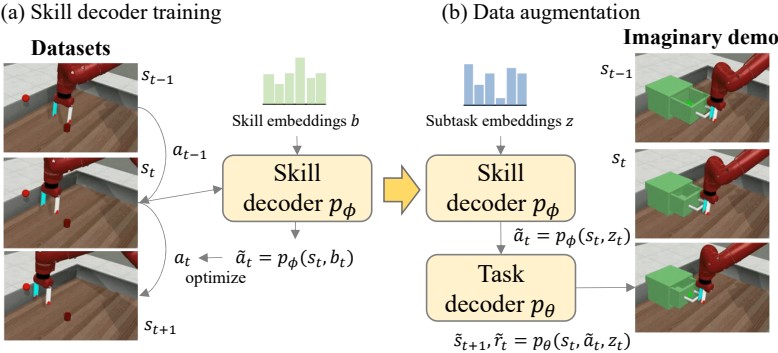

Figure 3: Data augmentation by imaginary demonstrations

where $\beta$ is a Lagrangian multiplier. By differentiating the right-hand side of (7) with respect to $q_\theta$ and following the optimal policy derivation procedure in [21, 22], we obtain the closed form solution that satisfies the return-weighted condition below.

$$q_\theta(\cdot|s) \propto \exp(\frac{1}{\beta}(\mathcal{R}^q_{s,\cdot} - V^q(s)))q(\cdot|s) \tag{8}$$

When omitting the baseline term $V^q(s)$ and up to the constant, we also obtain that the weighted skill regularized loss in (4) renders subtask embeddings to be matched with high-quality skills for a given task, thereby facilitating the task decomposition into shareable and achievable subtasks.

## 4 Data Augmentation by Imaginary Demonstrations

In offline RL, since a given static dataset might not fully represent its respective true MDP and further exploration is not allowed, it is common for RL agents to experience sub-optimal performance. Generative models and noise are used to generate additional trajectories and enable local exploration for offline RL algorithms without interaction. In this section, we introduce a data augmentation method specific to the aforementioned task decomposition with quality-aware skill regularization, so that we can tackle the overfitting and limited performance issue. While existing works aim at reducing the adverse effect of unseen states by exploiting state augmentation methods [23, 24, 25], we focus on augmenting such trajectories (imaginary demonstrations) that are likely to be generated by a high-quality skill-based learned policy.

Specifically, we integrate the task decoder $p_\theta$ and skill decoder $p_\phi$ into a generative model, as illustrated in Figure 3. Note that both $p_\theta$ and $p_\phi$ are established through the skill-regularized task decomposition. Then, we obtain

$$\tilde{a}_t, \ (\tilde{s}_{t+1}, \ \tilde{r}_t) = p_\phi(s_t, z_t), \ p_\theta(s_t, \tilde{a}_t, z_t) \text{ where } z_t = q_\theta(\tau_t) \tag{9}$$

where $q_\theta$ is the task embedding model and $\tau_t \sim \mathcal{D}$. Note that in this generative model, $p_\theta$ performs the same role of the world model in conventional model-based RL approaches.

The skill-regularized task decomposition enables $q_\theta$ to be an optimal policy in the MDP $\mathcal{M}^{p_\phi}$ as explained in (8). Accordingly, it turns out that the augmentation procedure in (9) yields a plausible trajectory similar to expert demonstrations, given that the high-quality skill corresponding to the trajectory is incorporated into $p_\phi$. In consequence, the procedure in (9) can alleviate the negative effect of low-quality datasets. Figure 4 depicts the effect of our data augmentation by imaginary demonstrations, where (a) and (b) show the difference of low-quality and high-quality datasets in terms of the state-action pair distribution. Our imaginary demonstrations in (c) generated from (a) the source dataset share similarity with (b) the expert dataset, while (d) the dataset generated by some conventional augmentation using Gaussian noise does not. The table shows the average reward of datasets calculated by reward relabeling, indicating that our skill-based augmentation by imaginary demonstrations produces higher quality data compared to the Gaussian noise-based augmentation.

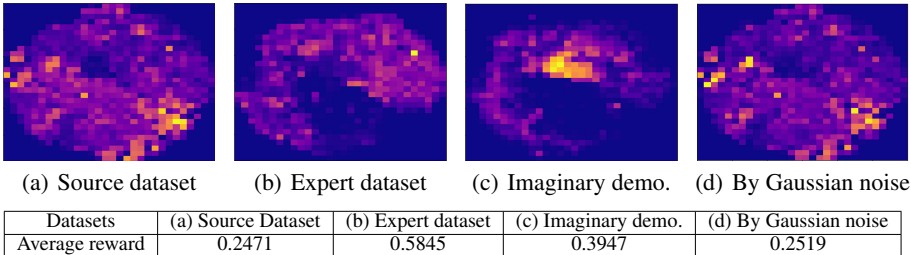

| Datasets | (a) Source Dataset | (b) Expert dataset | (c) Imaginary demo. | (d) By Gaussian noise |
|---|---|---|---|---|
| Average reward | 0.2471 | 0.5845 | 0.3947 | 0.2519 |

Figure 4: Examples of state-action pair distribution. (c) The imaginary demonstrations generated from (a) the source dataset look more similar to (b) the expert dataset than (a) the source dataset, while (d) the augmented dataset by Gaussian noise does not. The table lists the average rewards calculated by reward relabeling on the datasets in (a)-(d), respectively, illustrating the quality gain of (c) compared to (d) in terms of average rewards.

## 5  Experiments

In this section, we evaluate the performance of our model under various configurations of multi-task offline datasets.

**Experiment settings** For evaluation, we use several robotic manipulation tasks and drone navigation tasks using the Meta-world environment [26] and the Airsim drone simulator [27]. The detailed settings including hyperparameters and environment conditions can be found in Appendix.

**Comparison methods** We implement several multi-task RL methods for comparison including:

- TD3+BC [2] is a state-of-the-art offline RL algorithm, which incorporates a behavior cloning regularization term into the update steps of TD3, inducing the learned policy towards the actions found in the dataset. For multi-task learning, TD3+BC is extended to include one-hot encoded task representation as part of the state. We use this baseline to compare our approach with conventional multi-task RL algorithms without considering subtask decomposition.

- PCGrad [7] is a gradient surgery-based multi-task RL algorithm, which uses a projection function to remove the directional conflicts between gradients. This is implemented based on the hypothesis such that performance degradation can be exacerbated by gradient conflicts when training uncorrelated tasks.

- Soft modularization (SoftMod) [8] is a modular deep neural network architecture tailored for multi-task RL. To mitigate the negative impact of learning different tasks on a single policy, it leverages the softly weighted routing path on a set of modules that are specifically trained on multiple tasks. It also employs a loss-balancing strategy to rapidly adapt to the different learning progress of the tasks.

**Offline datasets** We create and use three types of datasets according to specific behavior policies at different quality levels. Medium-Replay (MR) denotes the datasets sampled by a learning process from the initial to partially-trained medium policies, Replay (RP) denotes the datasets sampled during a whole learning process, and Medium-Expert (ME) denotes the datasets sampled by a learning process from the medium to expert policies. Note that each dataset in MR, RP, and ME for a task contains episodic trajectories of 150, 100, and 50, respectively, unless otherwise stated.

### 5.1  Meta-world Tests

Here, we evaluate our model using the MT10 benchmark (i.e., 10 different control tasks) in Meta-world [26], where each task is given a specific manipulation objective such as opening a door or closing a window. The tasks share common primitive functions such as grasp and moving, so they can be seen as general multi-tasks with shared subtasks, which are consistent with our task decomposition strategy. The implementation detail can be found in Appendix.

**Performance on MT10 benchmark** Table 1 shows the performance in the average success rate on MT10 under mixed configurations of different datasets (MR, RP, ME). We compare the performance

| Datasets | | | Comparison | | | Our model | |
|---|---|---|---|---|---|---|---|
| MR | RP | ME | TD3+BC | PCGrad | SoftMod | SRTD | SRTD+ID |
| 10 | 0 | 0 | $19.73 \pm 2.71\%$ | $20.66 \pm 4.27\%$ | $13.43 \pm 2.67\%$ | $21.24 \pm 1.40\%$ | $\mathbf{23.87 \pm 2.22}\%$ |
| 0 | 10 | 0 | $25.93 \pm 5.91\%$ | $27.31 \pm 3.15\%$ | $29.04 \pm 0.58\%$ | $38.97 \pm 3.38\%$ | $\mathbf{41.91 \pm 5.88}\%$ |
| 0 | 0 | 10 | $23.93 \pm 3.99\%$ | $33.06 \pm 3.69\%$ | $39.61 \pm 1.02\%$ | $46.60 \pm 3.11\%$ | $\mathbf{49.29 \pm 3.35}\%$ |
| 7 | 0 | 3 | $22.13 \pm 1.05\%$ | $23.23 \pm 1.94\%$ | $20.80 \pm 4.97\%$ | $28.67 \pm 1.51\%$ | $\mathbf{32.53 \pm 4.90}\%$ |
| 5 | 3 | 2 | $25.13 \pm 1.49\%$ | $25.10 \pm 1.72\%$ | $28.73 \pm 0.56\%$ | $\mathbf{33.60 \pm 6.24}\%$ | $32.13 \pm 3.57\%$ |
| 5 | 0 | 5 | $16.53 \pm 4.71\%$ | $22.17 \pm 3.68\%$ | $28.13 \pm 4.59\%$ | $35.13 \pm 3.36\%$ | $\mathbf{36.80 \pm 5.27}\%$ |
| 4 | 3 | 3 | $27.60 \pm 3.25\%$ | $23.53 \pm 8.40\%$ | $25.87 \pm 1.26\%$ | $36.80 \pm 4.67\%$ | $\mathbf{43.53 \pm 3.32}\%$ |
| 3 | 0 | 7 | $23.53 \pm 1.98\%$ | $25.60 \pm 5.01\%$ | $31.77 \pm 3.52\%$ | $42.13 \pm 2.19\%$ | $\mathbf{44.93 \pm 5.35}\%$ |
| 0 | 7 | 3 | $24.93 \pm 2.44\%$ | $26.50 \pm 4.06\%$ | $30.33 \pm 1.31\%$ | $43.27 \pm 3.27\%$ | $\mathbf{43.73 \pm 3.88}\%$ |
| 0 | 5 | 5 | $24.70 \pm 3.99\%$ | $27.52 \pm 3.69\%$ | $32.06 \pm 1.02\%$ | $42.46 \pm 3.11\%$ | $\mathbf{44.42 \pm 3.35}\%$ |

Table 1: Performance on MT10 in the success rate with 95% confidence intervals (with 3 different random seeds). The Datasets column specifies the mixed configurations of different datasets; e.g., the row of (MR 10, RP 0, ME 0) corresponds to a specific configuration where each of all 10 task in MT10 has MR, and the row of (MR 5, RP 3, ME 2) corresponds to another mixed configuration where each of 5 tasks, 3 tasks, and 2 tasks has MR, RP, and ME, respectively.

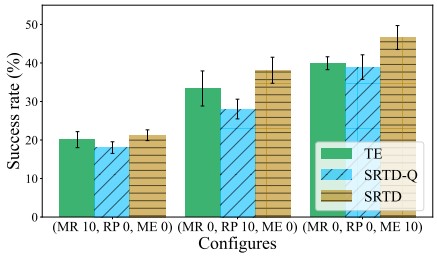

(a) Comparison of task embedding methods

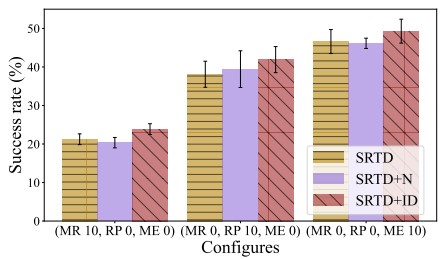

(b) Comparison of data augmentation methods

Figure 5: Effects of (a) quality-aware skill regularization and (b) imaginary demonstrations

achieved by our model (SRTD, SRTD+ID) and other methods (TD3+BC, PCGrad, SoftMod). Note that Skill Regularized Task Decomposition (SRTD) is trained as in Section 3, and SRTD with Imaginary Demonstrations (SRTD+ID) is trained by our whole model introduced in Sections 3 and 4. Our model (SRTD, SRTD+ID) yields the best performance consistently for all configurations, demonstrating its superiority with 8.67%~17.67% higher success rates, compared to the most competitive comparison method, SoftMod. SRTD consistently shows robust performance, while the comparison methods do not. TD3+BC and PCGrad show better performance for the configurations of low-quality datasets, e.g., the row of (MR 10, RP 0, ME 0), but SoftMod shows better performance for the configurations of high-quality datasets e.g., the row of (MR 0, RP 0, ME 10).

TD3+BC and PCGrad explore the orthogonality of tasks by accumulating task-specific knowledge separately without much interference when learning different tasks, and SoftMod rather exploits the commonality of the tasks by learning shared skills and dynamically extracting task-specific knowledge by the combination of its modules [7, 8]. Specifically, our TD3+BC implementation with one-hot task encoding tends to learn individual tasks separately, considering that the task encoding does not represent the semantic relation of different tasks explicitly. PCGrad intends for geometric separation of model updates for minimizing both the individual loss of each task and the distance between the weights of different tasks. SoftMod utilizes the shared weights of a modular network for jointly optimizing the multiple loss functions for the tasks [28]. Our SRTD tends to adjust both orthogonality and commonality using the quality-aware joint learning, so it can achieve robust performance for different mixed configurations. SRTD+ID improves the performance over SRTD by 2.97% at average, clarifying the benefit of imaginary demonstrations in offline datasets.

**Ablation study** Figure 5(a) shows the effect of our skill regularization, where TE denotes the task embedding without skill regularization (i.e., only $L_{TE}$ in (3) is used), and SRTD-Q denotes SRTD without the quality weighted term $\tilde{R}(s, a)$ in (4). SRTD shows better performance than the others consistently for all configurations. However, SRTD-Q shows worse performance than

| Datasets | | | Comparison | | | Our model | |
|---|---|---|---|---|---|---|---|
| MR | RP | ME | TD3+BC | PCGrad | SoftMod | SRTD | SRTD+ID |
| 6 | 0 | 0 | $12.71 \pm 2.27\%$ | $15.70 \pm 0.34\%$ | $16.76 \pm 3.58\%$ | $22.34 \pm 0.98\%$ | $\mathbf{24.60 \pm 2.25\%}$ |
| 0 | 6 | 0 | $13.06 \pm 2.93\%$ | $13.45 \pm 0.70\%$ | $21.19 \pm 1.98\%$ | $29.34 \pm 0.32\%$ | $\mathbf{30.77 \pm 2.16\%}$ |
| 0 | 0 | 6 | $14.82 \pm 1.99\%$ | $16.93 \pm 2.15\%$ | $26.35 \pm 2.35\%$ | $30.36 \pm 2.61\%$ | $\mathbf{35.83 \pm 0.80\%}$ |
| 2 | 2 | 2 | $14.66 \pm 2.55\%$ | $16.38 \pm 4.63\%$ | $24.07 \pm 2.28\%$ | $\mathbf{29.08 \pm 2.36\%}$ | $28.37 \pm 1.09\%$ |
| 1 | 2 | 3 | $13.23 \pm 0.47\%$ | $14.12 \pm 3.09\%$ | $22.80 \pm 1.37\%$ | $27.78 \pm 2.21\%$ | $\mathbf{34.18 \pm 1.16\%}$ |
| 3 | 2 | 1 | $12.18 \pm 2.14\%$ | $12.28 \pm 1.92\%$ | $18.67 \pm 2.89\%$ | $25.47 \pm 2.61\%$ | $\mathbf{27.51 \pm 1.79\%}$ |

Table 2: Performance on Airsim-based drone navigation in the normalized returns with 95% confidence intervals. Each return is normalized based on the maximum episodic return by a fully trained (online) RL agent that directly interacts with the environment. The Datasets column specifies the mixed configurations, as in Table 1, for 6 individual tasks in multi-task drone navigation.

TE, which specifies the benefit of quality-aware regularization. Figure 5(b) shows the effect of our data augmentation method, where SRTD+N denotes SRTD with the Gaussian noise-based data augmentation commonly used in offline RL [23]. We observe that SRTD+N rarely achieves performance gains, compared to SRTD. This clarifies the advantage of SRTD+ID that leverages high-quality skills to generate trajectories.

## 5.2   A Case Study for Airsim-based Drone Navigation

Here, we verify the applicability of our model in real-world problem scenarios by conducting a case study with autonomous quad-copter drones in the Airsim simulator [27]. We configure various realistic maps in PEDRA [29] and diverse wind patterns to build a multi-task drone flying environment. Regarding the RL formulation, the drone agent is set to observe its lidar data, position, speed, and angle of rotation, while it is set not to observe any data about wind patterns. With continuous observations, the drone agent conducts control actions by manipulating the 3-dimensional acceleration, and receives rewards based on the goal distance. For evaluation, we measure the normalized episodic return based on the maximum episodic return obtained by a fully trained online RL agent. Table 2 compares the performance by our model and the others. For all mixed configurations, our model shows the best performance, outperforming the competitive SoftMod by 5.01%∼11.37%.

## 6   Related Work

**Multi-task RL** Multi-task RL has been investigated for sample-efficiently dealing with complex control problems in real-world settings [30, 6, 8, 31]. By training a deep neural network with multiple tasks jointly, multi-task RL algorithms drive agents to learn how to share, reuse, and combine the knowledge across correlated tasks. Yang et al. [8] presented an explicit modular architecture with a soft routing network for training an integrated multi-task policy. This, called soft modularization, addresses the issue of unclear task relation in a single network such that which shared parameters are related to which tasks. Yu et al. [7] proposed a gradient surgery methodology that directly removes the negative effects of multi-task learning in a single policy, identifying and adjusting geometric conflicts of calculated gradients when learning different tasks.

**Task and skill embeddings in multi-task RL** Several approaches using task embeddings have been introduced in the context of meta RL [32, 33, 34], multi-task RL [6, 35], imitation learning [36, 37], and non-stationary RL [38, 39]. Pertsch et al. [40] demonstrated that with a pretrained low-level policy that readily achieves given skills, a high-level policy yielding appropriate skills can facilitate the learning efficiency, where skills are embedded in the latent space with expert data. Sodhani et al. [6] used additional metadata for learning a multi-task policy, exploiting task descriptions in natural language to represent the semantics and relation of tasks in the latent space. While these prior works rely on online interaction and they rarely consider heterogeneous datasets and different behavior polices, which are common in multi-task offline RL, our model employs the quality-aware regularization to handle the mixed configurations of multi-task datasets. We also devise a joint learning mechanism for skill and task representation in offline settings.

**Data augmentation in offline RL** To alleviate the issue of limited datasets and unseen states, several works exploited data augmentation [23, 41], data sharing [15, 11, 42], and model-based approach [43, 41] in offline RL. For example, Sinha et al. [23] tested several data augmentation

schemes, demonstrating the possible performance gain with offline RL algorithms. Yu et al. [11] presented the conservative Q-function that can judge which transitions are relevant for learning a specific task, thus establishing a conditional data sharing strategy upon data scarcity situations. Our data augmentation by imaginary demonstrations is in the same vein, but it focuses on exploiting common skills to generate trajectories that are likely to be generated by expert policies.

# 7   Conclusion

In this study, we proposed a novel multi-task offline RL model to tackle the problem of heterogeneous datasets of different behavior quality across tasks. In the model, skills and tasks are jointly learned with quality-aware regularization so that achievable subtasks are found and aligned with high-quality skills. Our model consistently yields robust performance upon various mixed configurations of different-quality datasets without requiring additional interaction with the environment. The direction of our future works is to investigate the hierarchy of skill representation with different temporal abstraction levels in multi-task offline RL. This will tackle the limitation of our model that considers only fixed-length sub-trajectories for task and skill embeddings.

# 8   Acknowledgement

We would like to thank anonymous reviewers for their valuable comments and suggestions. This work was supported by Institute of Information & communications Technology Planning & Evaluation (IITP) grant funded by the Korea government (MSIT) (No. 2022-0-00688, 2022-0-01045, 2022-0-00043, 2019-0-00421) and by the National Research Foundation of Korea (NRF) grant funded by MSIT (No. 2020M3C1C2A01080819).

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
