# OpenReview forum: "Skills Regularized Task Decomposition for Multi-task Offline Reinforcement Learning"
_NeurIPS.cc/2022/Conference — NeurIPS 2022 Accept_

### Official Review · Reviewer_ttxU · 2022-07-08

**Rating:** 7
**Confidence:** 4
**Soundness:** 4 excellent
**Presentation:** 3 good
**Contribution:** 3 good

**Summary:**

The authors present a method for multitask offline RL called skill regularized task decomposition. The authors learn autoencoders that embed skills (sequence of actions) and tasks (transition and reward models) into the same latent space, with greater weight given to higher-reward transitions. The authors also use these autoencoders to generate artificial demonstrations to augment the training dataset. In a 10-task Meta-world offline RL setup, the authors show that their method improves upon prior algorithms (TD3+BC, PCGrad, and SoftMod) for various different offline data mixture compositions.

**Questions:**

It’s not clear to me exactly how the TD3+BC baseline was trained. TD3+BC is a single-task algorithm, but appendix line 45 seems to say that it uses a “one-hot task encoding.” Are you considering a version of TD3+BC that has no knowledge of different tasks? Or are you one-hot encoding the tasks? It would be helpful if the paper clarified this.

**Limitations:**

I don’t see any negative societal implications that the authors failed to address.

**Strengths And Weaknesses:**

## Strengths
The authors compare against strong baselines: TD3+BC (a state-of-the-art single-task offline RL method) and two multitask RL methods (PCGrad and Soft modularization). The authors also test a variety of different offline dataset compositions, changing the proportion of data that is medium-replay (MR), replay (RP), and medium-expert (ME).

The authors split up their method into SRTD and SRTD+ID. This is helpful to help understand how much of the benefit comes from Skills Regularized Task Decomposition and how much comes from Imaginary Demonstrations.

Overall, I think multitask offline RL is relevant to the community. The authors give clear exposition for a new method and show that it outperforms strong baselines from prior work.

## Weaknesses
It is only in the appendix that the authors actually describe how they train an RL agent on top of the SRTD(+ID) embeddings. It would be helpful if the authors could fit this into the main text.

The authors don’t motivate MT10. Why are the tasks in MT10 interesting? It would be helpful if the authors included in the main text more qualitative information about the MT10 tasks and why they are interesting.

At line 243, the authors write: “It is because TD3+BC and PCGrad explore the orthogonality of tasks and SoftMod exploits the commonality of tasks.” I don’t fully understand this explanation and think it would be helpful if the authors elaborated on it.

---

> ### Author Response · Authors · 2022-08-02
> **Response to Reviewer ttxU (Part 1)**
>
> We thank the reviewer for the detailed comments and valuable feedback.
> ### Reviewer comment
> W1. It is only in the appendix that the authors actually describe how they train an RL agent on top of the SRTD(+ID) embeddings. It would be helpful if the authors could fit this into the main text.
> ### Author response
> We clarify that task and skill embeddings are later used for training an offline RL agent by adding the following statement in Section 2.2 (lines 102-103) of the main paper.
>
> "Then, using the subtask decomposition and augmented datasets with imaginary demonstrations, a multi-task policy is learned via offline RL algorithms."
>
>  We also revise Figure 1 to clearly depict the entire learning procedure including the skill regularized task decomposition, data augmentation by imaginary demonstration, and offline RL training.
>
> ### Reviewer comment
> W2. The authors don’t motivate MT10. Why are the tasks in MT10 interesting? It would be helpful if the authors included in the main text more qualitative information about the MT10 tasks and why they are interesting.
> ### Author response
> The MT10 benchmarks evaluate the performance by learning 10 different manipulation tasks in the meta-world environment such as reach, push, pick and place, open door, open drawer, close drawer, press button top-down, insert peg side, open window, and open box. It is one of the commonly used multi-task RL environments [1][2].
>
> In general, the tasks in MT10 share common functions such as grasp and moving, so they can be seen as general multi-tasks with shared subtasks. Specifically, each episode is set to have an individual configuration from 50 different positions of an object and goal, and each task is given a specific objective such as opening a door or closing a window. As such, MT10 allows for various experiment settings specific to multi-task RL with shared substasks, which is consistent with our strategy of skill-based task decomposition into achievable shared subtasks.
>
> The below is added in lines 226-230 of the main paper.
>
> "Here, we evaluate our model using the MT10 benchmark (i.e., 10 different control tasks) in Meta-world [26], where each task is given a specific manipulation objective such as opening a door or closing a window. The tasks share common primitive functions such as grasp and moving, so they can be seen as general multi-tasks with shared subtasks, which are consistent with our task decomposition strategy."
>
> [1] Yang, Ruihan, et al. "Multi-task reinforcement learning with soft modularization." Advances in Neural Information Processing Systems 33 (2020): 4767-4777.
>
> [2] Sodhani, Shagun, Amy Zhang, and Joelle Pineau. "Multi-task reinforcement learning with context-based representations." International Conference on Machine Learning. PMLR, 2021.
>
> ### Reviewer comment
> W3. At line 243, the authors write: “It is because TD3+BC and PCGrad explore the orthogonality of tasks and SoftMod exploits the commonality of tasks.” I don’t fully understand this explanation and think it would be helpful if the authors elaborated on it.
> ### Author response
> In general, multi-task learning approaches can be categorized into two groups: hard parameter sharing and soft parameter-sharing, as explained in Section 1 of [1].
> The former uses the shared weights of a model to jointly minimize multiple loss functions for multi-tasks, while the latter uses the separate weights of a model to minimize both the individual loss of each task and the distance between the model weights of different tasks.
> Overall, TD3+BC and PCGrad can be seen as the methods based on soft parameter sharing, and both are intended to establish the knowledge of multi-tasks without much interference by learning different tasks. Unlike those, soft modularization is a hard parameter sharing method, learning the common skills of multi-tasks and how to combine them on a modular network. All the learned knowledge about multi-tasks is represented in a shared modular network, but the task-specific knowledge is dynamically extracted by the combination of its modules.
>
> We revise the following sentence in the lines 241-243 of the main paper.
>
> "TD3+BC and PCGrad explore the orthogonality of tasks by accumulating task-specific knowledge separately and SoftMod exploits the commonality of tasks by learning shared skills across tasks."
>
> [1] Crawshaw, Michael. "Multi-task learning with deep neural networks: A survey." arXiv preprint arXiv:2009.09796 (2020).
>
> [2] Sodhani, Shagun, Amy Zhang, and Joelle Pineau. "Multi-task reinforcement learning with context-based representations." International Conference on Machine Learning. PMLR, 2021.

---

> > ### Comment · Reviewer_ttxU · 2022-08-04
> > **Line 243 Explanation Could Still be More Detailed**
> >
> > Thanks for the reply! I still think it would help make your paper more clear if you included a longer explanation about your claim at line 243 that "It is because TD3+BC and PCGrad explore the orthogonality of tasks and SoftMod exploits the commonality of tasks." Your proposed addition is helpful, but it would be even more helpful if you added something longer with more explanation.

---

> > > ### Author Response · Authors · 2022-08-05
> > > **Thanks for your detailed feedback!**
> > >
> > > Thanks for your detailed feedback! We add more explanations to the main paper for better understanding.
> > >
> > > We add the following sentence in lines 240-246 of the main paper.
> > >
> > > "TD3+BC and PCGrad focus on spatial isolation of model weights for minimizing both the individual loss of each task and the distance between the weights of different tasks, while SoftMod utilizes the shared weights of a modular network for jointly optimizing the multiple loss functions for the tasks [28].
> > > That is, TD3+BC and PCGrad explore the orthogonality of tasks by accumulating task-specific knowledge separately without much interference when learning different tasks, and SoftMod rather exploits the commonality of the tasks by learning shared skills and dynamically extracting task-specific knowledge by the combination of its modules [7, 8]."

---

> > > > ### Comment · Reviewer_ttxU · 2022-08-05
> > > > **Why is TD3+BC "Isolat[ing] Model Weights"?**
> > > >
> > > > Thanks for adding that longer discussion to the main paper, it helps me understand better what you are trying to say.
> > > >
> > > > Could you explain more why TD3+BC uses spatial isolation of model weights? If TD3+BC is just adding to the state a one-hot vector encoding the tasks, then couldn't many of the weights be shared across the different tasks?

---

> > > > > ### Author Response · Authors · 2022-08-06
> > > > > **Response to Reviewer ttxU (Why is TD3+BC "Isolat[ing] Model Weights"?)**
> > > > >
> > > > > Thank you for your thoughtful comment and insight.
> > > > > ### Reviewer comment
> > > > > Could you explain more why TD3+BC uses spatial isolation of model weights? If TD3+BC is just adding to the state a one-hot vector encoding the tasks, then couldn't many of the weights be shared across the different tasks?
> > > > >
> > > > > ### Author response
> > > > > Yes, many weights in a model network can be shared across tasks by TD3+BC. In our previous response, the spatial isolation by TD3+BC and PCGrad should have been explained more clearly with the commonality of multi-task knowledge. As the reviewer commented, when one-hot encoded task representation is used as input, TD3+BC exploits both sharing and separation of model weights. Given a one-hot vector as the input state for each task, its directly connected weights are associated with zero-inputs, but other weights tend to be shared across different tasks. However, this sharing by one-hot task encoding rarely implies that TD3+BC learns common (shared) knowledge across tasks because the encoding itself does not contain any semantic relation of different tasks [1, 2]. Accordingly, we consider that TD3+BC learns tasks by accumulating task-specific knowledge separately on all weights, consistent with soft parameter sharing methods. In that sense, PCGrad adjusts gradient updates and removes geometrical interference by different tasks, tending to focus on spatial isolation.
> > > > >
> > > > > We revise the following sentence in lines 243 - 248 of the main paper to clarify this explanation.
> > > > >
> > > > > "Specifically, our TD3+BC implementation with one-hot task encoding tends to learn individual tasks separately, considering that the task encoding does not represent the semantic relation of different tasks explicitly. PCGrad intends for geometric separation of model updates for minimizing both the individual loss of each task and the distance between the weights of different tasks. SoftMod utilizes the shared weights of a modular network for jointly optimizing the multiple loss functions for the tasks [28]."

---

> ### Author Response · Authors · 2022-08-02
> **Response to Reviewer ttxU (Part 2)**
>
>
> ### Reviewer comment
> Q1. It's not clear to me exactly how the TD3+BC baseline was trained. TD3+BC is a single-task algorithm, but appendix line 45 seems to say that it uses a “one-hot task encoding.” Are you considering a version of TD3+BC that has no knowledge of different tasks? Or are you one-hot encoding the tasks? It would be helpful if the paper clarified this.
> ### Author response
> We adapt TD3+BC to have the one-hot encoded vector for multi-tasks in the RL input state.
> For example, consider two different tasks such as window open and window close that are learned by TD3+BC; the one-hot [1, 0] and [0, 1] can be added in the state for window open and window close tasks, respectively.
> This approach is commonly used to represent multiple MDPs as a single MDP,
> \begin{equation}
>     {(\mathcal{S}, \mathcal{A}, \mathcal{P}, R)}_{i\in\mathcal{I}} \rightarrow (\mathcal{S} \times \mathcal{I}, \mathcal{A}, \mathcal{P}, R).
> \end{equation}
> We add the following sentence in the lines 207-209 of the main paper.
>
> "For multi-task learning, TD3+BC is extended to include one-hot encoded task representation as part of the state. We use this baseline to compare our approach with conventional multi-task RL algorithms without considering subtask decomposition. "

---

### Official Review · Reviewer_5BuZ · 2022-07-08

**Rating:** 7
**Confidence:** 5
**Soundness:** 3 good
**Presentation:** 3 good
**Contribution:** 3 good

**Summary:**

This paper proposes a skill-based multi-task RL technique to learn the shareable knowledge across multiple offline datasets that are generated by behavior policies of different quality. Specifically, they develop a task decomposition method where the Wasserstein auto-encoder (WAE) is used to represent both skills and tasks in the same latent space. Then they use the quality-weighted loss as a regularization term to induce tasks to be decomposed into subtasks that are more consistent with high-quality skills than others. To further improve the performance, they augment datasets with imaginary trajectories relevant to high-quality skills for each task. They evaluate their method on two types of Offline RL tasks, manipulation tasks and drone navigation tasks, and the results show that their method outperforms other state-of-the-art algorithms.

**Questions:**

* What does the red region mean in Figure 1?

* What’s the advantage of generating imaginary data using skill embedding? It is hard to tell that the imaginary trajectories are more like high-quality datasets in Figure 4. It would be better if the authors can compare their method with a baseline that uses normal counterfactual data augmentation methods.

* I notice that the three baselines used in the experiments are not designed for tackling the Multi-task Offline RL problems and are published in 2020. Since I am not familiar with SOTA algorithms in this field, I can’t recommend suitable baselines. But if there exist similar methods, please add more baselines to make the comparison fair.


**Ethics Review Area:**

["I don’t know"]

**Limitations:**

Limitations are not fully discussed in the paper. One limitation is that the application is limited if the skills and tasks are complicated and can’t be represented only by state-action pairs.

**Strengths And Weaknesses:**

### Strengths

* This paper has a clear motivation for solving the multi-task problem in the Offline RL setting.
* The proposed method is complicated but novel.
* The paper is well-written and easy to follow.

### Weaknesses
* Some figures are not intuitive enough and do not provide enough information. For example, Figure 1 and the right part of Figure 2.
* There might be some missing baselines in the experiment part.

---

> ### Author Response · Authors · 2022-08-02
> **Response to Reviewer 5BuZ  (Part 1)**
>
> We thank the reviewer for the detailed comments and valuable feedback.
> ### Reviewer comment
> W1. Some figures are not intuitive enough and do not provide enough information. For example, Figure 1 and the right part of Figure 2.
> ### Author response
> In order to emphasize the embedding pattern difference in the latent space for low-quality and high-quality sub-trajectories, which are achieved by our skill regularized task decomposition, we revise Figure 1 with more clear separation between two dotted circles. The circles correspond to respective task embedding procedures for low-quality and high-quality sub-trajectories. We also add several legends for better understanding.
>
> On the right part of Figure 2, to clearly describe how each task embedding locates around its associated skill embedding from the same sub-trajectory, we minimize the overlapping region of two circles that correspond to different task embeddings, and add several legends that indicate the task embeddings.
>
> ### Reivewer comment
> W2. There might be some missing baselines in the experiment part.
>
> ### Author response
> We add more experiments for comparing the quality of augmented data generated by different augmentation strategies.
> We also add experiments with additional baselines in multi-task offline RL such as CDS (conservative data sharing) in [1].
> More details are in the response to Q2 and Q3 below.
>
> [1] Yu, Tianhe, et al. "Conservative data sharing for multi-task offline reinforcement learning." Advances in Neural Information Processing Systems 34 (2021): 11501-11516.
>
> ### Reviewer comment
> Q1. What does the red region mean in Figure 1?
> ### Author response
> In the main paper, we revise Figure 1 and add several legends for more clear description.  In Figure 1, the latent space $\mathcal{Z}$ is characterized by different embedding patterns depending on the quality of sub-trajectories, for which episodic returns are used. The red-colored dotted circle (red region) represents an embedding example when the return of sub-trajectory $\tau_N$ is small, and the green-colored dotted circle represents the opposite example when the return is large. Specifically, in the red-colored dotted circle region, embedding $z_N$ of sub-trajectory $\tau_N$ of task N is located as $z'_N$ further from its corresponding low-quality skill $b_N$ (the action sequence of the sub-trajectory $\tau_N$ with small returns). In contrast, in the green-colored dotted region, embedding $z_1$ of a sub-trajectory $\tau_1$ of task 1 is located as $z'_1$ closer to its corresponding high-quality skill $b_1$ (the action sequence of the sub-trajectory $\tau_1$ with large returns).
>
> ### Reviewer comment
> Q2. What’s the advantage of generating imaginary data using skill embedding?
> ### Author response
> Imaginary data augmentation using skill embeddings is intended to efficiently produce trajectories similar to expert demonstrations.
> The reason why we can emulate such expert demonstrations is that as proved in Section 3.2 of the main paper, the task encoder $q_{\theta}$ learned by skill-regularized task decomposition behaviors similar to the optimal policy that is induced by the MDP $\mathcal{M}^{p_{\phi}}$, in which the skill decoder $p_{\phi}$ transforms a high-level action (i.e., task embedding) to a sequence of low-level actions. As such, this skill-based imaginary demonstration tends to alleviate the negative effect of low-quality datasets.
>
> The below table shows the average reward of datasets calculated by reward relabeling, indicating that our skill-based augmentation by imaginary demonstrations produces higher quality data, compared to the Gaussian noise-based augmentation that is commonly used in offline RL [1]. We add the table and the above statement in Section 4 of the main paper.
> Furthermore, we consider the augmentation method by Gaussian noise as a counterfactual experiment to our skill-based augmentation, and we provide their comparison in the ablation study (Figure 5(b)) of Section 5.1 in the main paper.
>
> |              Datasets              | Average reward |
> |:----------------------------------:|:--------------:|
> | Source data (without augmentation) |     0.2471     |
> |            Expert data             |     0.5845     |
> |      Imaginary demonstrations      |     0.3947     |
> |  Source data with Gaussian noise   |     0.2519     |
>
> [1] Sinha, Samarth, Ajay Mandlekar, and Animesh Garg. "S4RL: Surprisingly simple self-supervision for offline reinforcement learning in robotics." Conference on Robot Learning. PMLR, 2022.

---

> ### Author Response · Authors · 2022-08-02
> **Response to Reviewer 5BuZ (Part 2)**
>
>
> ### Reviewer comment
> Q3. notice that the three baselines used in the experiments are not designed for tackling the Multi-task Offline RL problems and are published in 2020. Since I am not familiar with SOTA algorithms in this field, I can’t recommend suitable baselines. But if there exist similar methods, please add more baselines to make the comparison fair.
> ### Author response
> For baselines, we use TD3+BC, PCGrad, and Soft modularization methods; although these methods did not directly tackle the multi-task offline RL problem in prior works, they have been adapted and extended for the problem in our work.
> TD3+BC is a general single-task offline RL algorithm. For handling multi-task settings, it is extended with one-hot task encoding where each one-hot corresponds to an individual task in multi-task conditions and it is contained in the RL state representation. This one-hot encoded task representation is commonly used in general multi-task learning environments [1][2].
> PCGrad is a multi-task (online) RL algorithm, aiming at removing the interference between different learning tasks through gradient modulation schemes. This algorithm is adapted for multi-task offline RL, combining with gradient-based offline RL algorithms such as TD3+BC.
> Soft modularization is a multi-task RL method that exploits a modular policy network and soft routing structure. We use this method by adopting offline RL algorithms (i.e., CQL) when training both the modular and routing networks.
>
> There are very few research works that directly address the problem of multi-task offline RL, and even they have some specific assumption on RL formulation such that the reward function is known [3][4] or data sampling can be conducted again through interaction with the environment [4].
> In conservative data sharing (CDS) [3], the data limitation problem in offline RL was discussed and selective data sharing strategies across different task datasets were presented.
> Unlike CDS, we don't assume that reward function for each tasks is known, so we compare our model and CDS under different experiment conditions, where CDS exploits known reward functions but our model does not. We observe that CDS achieves good performance when high-quality data is sufficiently given but its performance much degrades when high-quality data is not sufficiently given. We speculate that it is because CDS shares only the transitions with high Q-values learned by CQL algorithm. In the table, the dataset configurations (MR 10, RP 0, ME 0), (MR 0, RP 10, ME 0), and (MR 5, RP3, ME 2) represent relatively low-quality conditions, while the dataset configurations (MR 0, RP 0, ME 10) and (MR 4, RP 3, ME 3) represent relatively high-quality conditions. For the former configurations, we observe better performance by SRTD+ID, and for the latter configurations, we observe comparable performance between SRTD+ID and CDS.
>
>
> | Datasets         | SRTD+ID      | CDS  |
> |------------------|--------------|------|
> |(MR 10, RP 0, ME 0)| 23.87 ± 2.22% | 17.50 ± 2.10% |
> |(MR 5, RP 3, ME 2)| 32.13 ± 3.57% | 29.60 ± 3.30% |
> |(MR 0, RP 10, ME 0)| 41.91 ± 5.88%| 35.88 ± 2.14% |
> |(MR 4, RP 3, ME 3)| 43.53 ± 3.32% | 42.17 ± 2.57% |
> |(MR 0, RP 0, ME 10)| 49.29 ± 3.35 \% | 48.12 ± 1.41% |
>
>
> We add these results in Appendix.
>
> [1] Yu, Tianhe, et al. "Meta-world: A benchmark and evaluation for multi-task and meta reinforcement learning." Conference on robot learning. PMLR, 2020.
>
> [2] Sodhani, Shagun, Amy Zhang, and Joelle Pineau. "Multi-task reinforcement learning with context-based representations." International Conference on Machine Learning. PMLR, 2021.
>
> [3] Yu, Tianhe, et al. "Conservative data sharing for multi-task offline reinforcement learning." Advances in Neural Information Processing Systems 34 (2021): 11501-11516.
>
> [4] Yarats, Denis, et al. "Don't Change the Algorithm, Change the Data: Exploratory Data for Offline Reinforcement Learning." arXiv preprint arXiv:2201.13425 (2022).
>
> ### About limitation
> We revise the conclusion to include our limitation on variable lengths of sub-trajectories for task embedding and represent our plan to tackle the limitation.

---

> > ### Comment · Reviewer_5BuZ · 2022-08-05
> > **Thanks for the response**
> >
> > Thanks for the authors' response. Most of my questions have been answered. After doing some literature review, I also realized that the multi-task offline RL topic is not widely investigated. Thus, I agree with the baseline selection in the paper, which adapts single-task Offline RL methods or multi-task online methods. I raised my score to 7 to recommend the acceptance of this paper.

---

> ### Comment · Reviewer_ttxU · 2022-08-04
> **TD3+BC is a good SOTA Baseline**
>
> As someone with experience in offline RL, I think TD3+BC is a good SOTA baseline. It is a simple algorithm with strong performance across many tasks. See, for example:
>
> Offline Reinforcement Learning with Implicit Q-Learning, ICLR 2022. https://arxiv.org/abs/2110.06169
>
> RvS: What is Essential for Offline RL via Supervised Learning?, ICLR 2022. https://arxiv.org/abs/2112.10751

---

> > ### Comment · Reviewer_5BuZ · 2022-08-05
> > **Thanks for the suggestion.**
> >
> > Thank for reviewer ttxU's suggestion. I agree that TD3-BC could also be a strong baseline in the multi-task offline RL setting.

---

### Official Review · Reviewer_qkkL · 2022-07-08

**Rating:** 7
**Confidence:** 3
**Soundness:** 3 good
**Presentation:** 3 good
**Contribution:** 3 good

**Summary:**

This paper proposes an offline multi-task RL algorithm which introduces a novel skill-level and task-level encoder, which is used to encode subtrajectories from offline data as well as tasks into a shared latent space. This generative model can be used to generate imaginary data for improved training. The method is tested on simulated Metaworld environments (MT-10) as well as a drone navigation task in the Airsim simulator and shows improved performance compared to prior works TD3+BC, PCGrad, and SoftMod.

**Questions:**

- I don’t quite understand the difference between TE and SRTD-Q in the ablation study section of the paper. The paper says TE is the task embedding without skill regularization (only $L_{TE}$ is used), but isn’t the only additional term the quality-aware term from Equation (4) (which is what SRTD-Q removes?) I would appreciate a clarification.
- I can’t find any mention in the main paper that the learned subtask and subskill embeddings are later used in training an offline RL agent using TD3+BC, and had to look in the appendix for this – I strongly recommend this be included in the main text for clarity.


**Limitations:**

The limitations of the method don't appear to be explicitly discussed, although a direction of future work is discussed. I would encourage the authors to address the current limitations and potential negative societal impact of the work.

**Strengths And Weaknesses:**

Strengths:
- The skill and task encoding strategy is very novel to me and has appealing properties such as encoding skills and tasks into the same latent space, where the distance between skills and tasks is meaningful.
- The experimental evaluation compares to sensible baselines (TD3+BC, PCGrad, SoftMod) and achieves strong results. It’s nice that the methods are compared across different configurations for the dataset quality as well, and demonstrates that the SRTD method is able to handle cases with lower quality data as well as higher-quality datasets.
- The ability to generate additional imaginary high-quality data is a nice property of the formulation, and Figure 4 in the paper is a nice qualitative result in this direction. However, the additional imaginary data doesn’t improve performance as much as I may have expected when looking at Figure 5b, but does seem to help a bit. Based on the ablations, It seems that most of the performance improvements compared to the TD3+BC baseline are coming from using a high level skill embedding at all, which is also an interesting result (but please let me know if I am misinterpreting the results).

Weaknesses:
- One weakness of the current formulation is that the subtrajectories and skill embeddings represent sequences of states and actions with a fixed time length. How is the subtask length chosen, and how are the subtrajectories sampled during training of the WVAEs?
- While the quality-aware skill regularization seems like a reasonable heuristic, it seems like estimating the quality of a particular sub-trajectory based on the episodic returns is likely error-prone if a trajectory is not of uniform quality. This could work well in settings where trajectories are collected such that they contain either solely low-quality or high-quality behaviors such as the ones examined in this work, but may not be able to pick good behaviors from mixed quality data.

---

> ### Author Response · Authors · 2022-08-02
> **Response to Reviewer qkkL (Part 1)**
>
> We thank the reviewer for the detailed comments and valuable feedback.
> ### Reviewer comment
> W1. One weakness of the current formulation is that the sub-trajectories and skill embeddings represent sequences of states and actions with a fixed time length. How is the subtask length chosen, and how are the sub-trajectories sampled during training of the WVAEs?
> ### Author response
> To estimate the quality of sub-trajectories, our proposed method uses unbiased quality measures such as advantage or episodic return. In the case of having sub-trajectories of variable lengths, the quality measure might vary depending on their length. While it is also interesting to investigate how to stably approximate the quality in variable length settings, we define the length of sub-trajectories as a fixed hyperparameter and perform experiments with various dataset quality conditions, focusing on the quality-aware skill regularization.
>
> We use $n$-length sub-trajectories $\tau_{t-n : t}$ in the task embedding procedure, and we use $2n$-length sub-trajectories $\tau_{t-n : t+n-1}$ in the skill embedding procedure.
> In our implementation, task embeddings (generated by the task encoder $q_{\theta}$ in Figure 2 of the main paper) are used as input for a learned RL policy, so only $n$-length sub-trajectories (without future transitions) are used, similar to the task embedding method in [1].
> However, sub-trajectories for skill embeddings are $2n$-length transitions including the past of $n$-length and the future of $n$-length, since skills abstract the action sequence conditioned on a given (current) state, similar to the skill embedding method in [2]. The table below shows the performance in multi-task success rates for MT10 achieved by different sub-trajectory length settings $n = 2, 4, 8, 16$. As shown, no significant difference in performance is observed as long as $n$ is not too short.
>
> | Datasets            | n=2           | n=4           | n=8           | n=16          |
> |---------------------|---------------|---------------|---------------|---------------|
> | (MR 10, RP 0, ME 0) | 19.75 ± 1.01% | 21.24 ± 1.40% | 20.28 ± 1.25% | 21.51 ± 2.25% |
> | (MR 0, RP 10, ME 0) | 34.32 ± 2.12% | 38.97 ± 3.38% | 38.50 ± 3.58% | 40.64 ± 6.25% |
> | (MR 0, RP 0, ME 10) | 38.52 ± 3.44% | 46.60 ± 3.31% | 46.43 ± 2.81% | 44.21 ± 4.84% |
>
> When training the WAEs, in terms of sampling methods, we compare two different approaches, quality-weighted random sampling and uniformly random sampling. It is observed that the quality-weighted random sampling often degrades the performance of learned RL policies, as it tends to concentrate on specific patterns of sub-trajectories, thus restricting the transition coverage in the environment. Hence, we use the uniformly random sampling method, considering that our task embedding relies on the Markovian property of the environment where the coverage of transitions sampled from offline data is important [3].
>
> |       Datasets        |  Uniform sampling  |  Quality weighted sampling  |
> |:---------------------:|:------------------:|:---------------------------:|
> |  (MR 0, RP 10, ME 0)  |   38.97 ± 3.38%    |       24.85 ± 7.14 %        |
>
> We add these results in Appendix. We will be able to include more experiment results with various dataset conditions for the above tables in the final paper.
>
> [1] Rakelly, Kate, et al. "Efficient off-policy meta-reinforcement learning via probabilistic context variables." International conference on machine learning. PMLR, 2019.
>
> [2] Nam, Taewook, et al. "Skill-based Meta-Reinforcement Learning." arXiv preprint arXiv:2204.11828 (2022).
>
> [3] Wang, Ruosong, Dean P. Foster, and Sham M. Kakade. "What are the statistical limits of offline RL with linear function approximation?." arXiv preprint arXiv:2010.11895 (2020).

---

> > ### Comment · Reviewer_qkkL · 2022-08-05
> > **Thank you for the response**
> >
> > Thank you for the detailed response to my review. I appreciate the clarifications about the setup of the ablation study and the modifications the authors have made to the text to improve clarity. The experiments showing that the method is relatively robust to skill selection length are also convincing in showing that the performance is not significantly tied to that hyperparameter. The demonstration of HCA as quality estimator is also helpful! I have increased my score accordingly.

---

> ### Author Response · Authors · 2022-08-02
> **Response to Reviewer qkkL (Part 2)**
>
> ### Reviewer comment
> W2. While the quality-aware skill regularization seems like a reasonable heuristic, it seems like estimating the quality of a particular sub-trajectory based on the episodic returns is likely error-prone if a trajectory is not of uniform quality. This could work well in settings where trajectories are collected such that they contain either solely low-quality or high-quality behaviors such as the ones examined in this work, but may not be able to pick good behaviors from mixed quality data.
>
> ### Author response
>
> While we use episodic returns for quality estimation and sub-trajectory relabeling, our SRTD can be readily extended to other quality-estimation methods. For example, Hindsight credit assignment (HCA) [1] can be used for quality estimation and sub-trajectory relabeling in SRTD, where HCA exploits the advantage for hindsight relabeling, i.e.,
> \begin{equation}
>     A_\pi(s, a) = \mathbb{E}_{\tau\sim\mathcal{D}}\left[ \left( 1 - \frac{\pi(a|s)}{\pi_z(a|s, \tilde{R}(s, a))}\right) * \tilde{R}(s, a)\right]
> \end{equation}
> where $\tilde{R}(s, a)$ is a return and $\pi_z$ is a return conditioned policy.
> Compared to the case of using the episodic returns in SRTD, our experiments rarely specify any performance improvement (i.e., as shown in the first 3 rows in the below table).
> That was expected to some extent because sampled transitions within an episode (or trajectory) turn out to be relatively either uniformly low-quality or high-quality in our datasets. In the offline RL context, it is common for offline dataset collection that a behavior (sampling) policy remains the same during an episode as it is learned [2].
>
> We also test the other case, the mixed-quality within an episode (MIX-EPI) where the behavior policy's quality is frequently changed even during a single episode. We deliberately set a sequence of  sampling policies for each episode of MIX-EPI datasets such that different policies are used for a few timesteps in rotation. This data collection emulates the environment where the quality of sub-trajectories is highly variable within each individual episode.
> In the MIX-EPI 10 case of the below table, we observe the performance difference achieved by SRTD and the SRTD variant with HCA (SRTD+HCA) for MT10; this motivates us as our future research to investigate other quality estimation and relabeling strategies for a wide range of mixed configurations of different quality datasets.
> The experiment results including SRTD with HCA for MIX-EPI datasets are added and discussed in Appendix.
>
> |      Datasets       |     SRTD      |  SRTD + HCA   |
> |:-------------------:|:-------------:|:-------------:|
> | (MR 10, RP 0, ME 0) | 21.24 ± 1.40% | 22.14 ± 1.09% |
> | (MR 0, RP 10, ME 0) | 38.97 ± 3.38% | 36.50 ± 2.01% |
> | (MR 0, RP 0, ME 10) | 46.60 ± 3.11% | 47.06 ± 2.18% |
> |    (MIX-EPI 10)     | 39.60 ± 3.24% | 42.4 ± 1.95 % |
>
> [1] Harutyunyan, Anna, et al. "Hindsight credit assignment." Advances in neural information processing systems 32 (2019).
>
> [2] Fu, Justin, et al. "D4rl: Datasets for deep data-driven reinforcement learning." arXiv preprint arXiv:2004.07219 (2020).

---

> ### Author Response · Authors · 2022-08-02
> **Response to Reviewer qkkL (Part 3)**
>
>
> ### Reviewer comment
> Q1. I don’t quite understand the difference between TE and SRTD-Q in the ablation study section of the paper. The paper says TE is the task embedding without skill regularization (only $L_{TE}$ is used), but isn’t the only additional term the quality-aware term from Equation (4) (which is what SRTD-Q removes?) I would appreciate a clarification.
>
> ### Author response
> The entire loss function of skill-regularized task decomposition is defined as
>
> \begin{equation}
> L_{SRTD}(\theta) = L_{TE}(\theta) + L_{SR}(\theta) + \lambda L_{PR}(\lbrace\tilde{z_i}\rbrace_{i=1}^m \sim P_Z, q_\theta(\lbrace \tau_{t_i}\rbrace_{i=1}^m))
> \end{equation}
>
> where $L_{TE}$ and $L_{SR}$ is defined as
>
> \begin{equation}
> L_{TE}(\theta) = \frac{1}{m}\sum_{i=1}^{m} \sum_{j=-n}^{0} \lVert (s_{t_i+j+1}, r_{t_i+j}) - p_\theta(s_{t_i+j}, a_{t_i+j}, q_\theta(\tau_{t_i})) \rVert_2,\ \
>  L_{SR} (\theta) = \frac{1}{m} \sum_{i=1}^m \tilde{R}(s_{t_i}, a_{t_i}) \cdot \lVert q_\theta(\tau_{t_i}) - q_\phi(d_{t_i}) \rVert_2.
> \end{equation}
>
> TE denotes the task embedding without skill regularization (i.e., only $L_{TE}$ is used), where its entire loss  defined as
> \begin{equation}
>     L_{TE}(\theta) + \lambda L_{PR}(\lbrace\tilde{z_i}\rbrace_{i=1}^m \sim P_Z, q_\theta(\lbrace\tau_{t_i}\rbrace_{i=1}^m)).
> \end{equation}
>
> SRTD-Q denotes SRTD without the quality-weighted term $\tilde{R}(s_{t_i}, a_{t_i})$, where its entire loss defined as
> \begin{equation}
> L_{TE}(\theta) + L_{SR-Q}(\theta) + \lambda L_{PR}(\lbrace\tilde{z_i}\rbrace_{i=1}^m \sim P_Z, q_\theta(\lbrace\tau_{t_i}\rbrace_{i=1}^m)) \ \text{where}\ \ L_{SR-Q} (\theta) = \frac{1}{m} \sum_{i=1}^m \lVert q_\theta(\tau_{t_i}) - q_\phi(d_{t_i}) \rVert_2.
> \end{equation}
>
> We add the following sentence in the lines 259-261 of the main paper.
>
> "Figure 5(a) shows the effect of our skill regularization, where TE denotes the task embedding without skill regularization (i.e., only $L_{TE}$ in (3) is used), and SRTD-Q denotes SRTD without the quality weighted term $\tilde{R}(s, a)$ in (4)."
>
> ### Reviewer comment
> Q2. I can’t find any mention in the main paper that the learned subtask and subskill embeddings are later used in training an offline RL agent using TD3+BC, and had to look in the appendix for this – I strongly recommend this be included in the main text for clarity.
>
> ### Author response
> We clarify that task and skill embeddings are later used for training an offline RL agent by adding the following statement in Section 2.2 (lines 102-103) of the main paper.
>
> "Then, using the subtask decomposition and augmented datasets with imaginary demonstrations, a multi-task policy is learned via offline RL algorithms."
>
> We also revise Figure 1 to clearly depict the entire learning procedure including the skill regularized task decomposition, data augmentation by imaginary demonstration, and offline RL training.
>
> ### About limitation
> We revise the conclusion to include our limitation on variable lengths of sub-trajectories for task embedding and represent our plan to tackle the limitation.

---

### Public Comment · ~Khoi_Hoang_Do1 · 2022-12-26
**Offline Data set**

Hi authors, I'm really interested in your research work. However, when I come into your supplementary material and try to download the offline_data.zip as shown as a gg drive link in the readme file, it's actually broken. Could you please give me the new link where I can download the data or show me how I achieved it in the material you supplied?
In fact, I have tried to generate data from your supplied code, but it only gives me the offline_data_random with a random policy. I have also checked whether I can generate the replay data but there is no promising result.

Hope to hear you soon

Sincere!

---

> ### Public Comment · ~Minjong_Yoo2 · 2022-12-26
> **fix download link**
>
> we fixed the download link in the supplementary material.
>
> Thanks for finding the problem.

---

> > ### Public Comment · ~Khoi_Hoang_Do1 · 2022-12-27
> > **Thank you**
> >
> > Thank you so so much

---

### Meta-Review · Area_Chair_yWhU · 2022-08-25

**Recommendation:** Accept
**Confidence:** Certain

**Metareview:**

The reviewers appreciated the authors' response and clarifications. Given the feedback from the reviewers and the discussion, I would like to recommend this paper for acceptance and congratulate them on a strong submission. I encourage the authors to address the reviewers' comments for the final version of the paper.

**Award:**

No

---

### Decision · Program_Chairs · 2022-09-14

Accept